# Transcriptome-Wide Analysis of microRNA–mRNA Correlations in Tissue Identifies microRNA Targeting Determinants

**DOI:** 10.3390/ncrna9010015

**Published:** 2023-02-13

**Authors:** Juan Manuel Trinidad-Barnech, Rafael Sebastián Fort, Guillermo Trinidad Barnech, Beatriz Garat, María Ana Duhagon

**Affiliations:** 1Sección Genómica Funcional, Facultad de Ciencias, UDELAR, Iguá 4225, Montevideo 11400, Uruguay; 2Departamento de Genética, Facultad de Medicina, Universidad de la República, Montevideo 11800, Uruguay; 3Departamento de Genómica, Instituto de Investigaciones Biológicas Clemente Estable, Av. Italia 3318, Montevideo 11600, Uruguay; 4Instituto de Computación, Facultad de Ingeniería, Universidad de la República, Montevideo 11300, Uruguay

**Keywords:** microRNA, 3′supplementary pairing, transcriptome, TCGA, offset, GC-content, correlation

## Abstract

MicroRNAs are small RNAs that regulate gene expression through complementary base pairing with their target mRNAs. A substantial understanding of microRNA target recognition and repression mechanisms has been reached using diverse empirical and bioinformatic approaches, primarily in vitro biochemical or cell culture perturbation settings. We sought to determine if rules of microRNA target efficacy could be inferred from extensive gene expression data of human tissues. A transcriptome-wide assessment of all the microRNA–mRNA canonical interactions’ efficacy was performed using a normalized Spearman correlation (Z-score) between the abundance of the transcripts in the PRAD-TCGA dataset tissues (RNA-seq mRNAs and small RNA-seq for microRNAs, 546 samples). Using the Z-score of correlation as a surrogate marker of microRNA target efficacy, we confirmed hallmarks of microRNAs, such as repression of their targets, the hierarchy of preference for gene regions (3′UTR > CDS > 5′UTR), and seed length (6 mer < 7 mer < 8 mer), as well as the contribution of the 3′-supplementary pairing at nucleotides 13–16 of the microRNA. Interactions mediated by 6 mer + supplementary showed similar inferred repression as 7 mer sites, suggesting that the 6 mer + supplementary sites may be relevant in vivo. However, aggregated 7 mer-A1 seeds appear more repressive than 7 mer-m8 seeds, while similar when pairing possibilities at the 3′-supplementary sites. We then examined the 3′-supplementary pairing using 39 microRNAs with Z-score-inferred repressive 3′-supplementary interactions. The approach was sensitive to the offset of the bridge between seed and 3′-supplementary pairing sites, and the pattern of offset-associated repression found supports previous findings. The 39 microRNAs with effective repressive 3′supplementary sites show low GC content at positions 13–16. Our study suggests that the transcriptome-wide analysis of microRNA–mRNA correlations may uncover hints of microRNA targeting determinants. Finally, we provide a bioinformatic tool to identify microRNA–mRNA candidate interactions based on the sequence complementarity of the seed and 3′-supplementary regions.

## 1. Introduction

MicroRNAs are small, single-stranded RNAs that shape the expression of messenger RNAs (mRNAs) and are essential in metazoans. Because of this, microRNAs influence all developmental processes and diseases, including cancer [1]. In microRNA targeting, the seed region (positions 2–7 or 2–8 of the microRNA) is the primary determinant of targeting efficacy and specificity, interacting with the 3′ untranslated region (UTR) or the coding sequence (CDS) of the target mRNAs [2,3,4,5,6]. Since the seed region is only composed of 6–7 contiguous nucleotides, numerous sites in the transcriptome are capable of base pairing with it, but not all are functionally associated with microRNA-mediated repression [7,8,9]. Although seminal studies have discovered the main rules governing microRNA–mRNA interaction, several aspects still remain unsolved, and the identification of biologically relevant interactions is difficult [1].

Mounting evidence demonstrated that nucleotides outside of the seed of the microRNA contribute to target recognition through complementary base pairing [2,3,4,5,10,11,12,13,14,15]. The systematic analysis of microRNA pairing likelihood revealed the conservation of nucleotides 13–16 [2]. However, this 3′-supplementary pairing has shown little influence on on-site affinity and efficacy [2,12,15], and actually, after purifying selection, only approximately 5% of the seed-matched regulatory sites appear to involve it [16]. More recent crystallography of human Ago2 complexed to microRNA-122 and a target RNA uncovered microRNA–mRNA interactions through the 2–8 seed and also the 3′-supplementary region of the microRNA (nucleotides 13–16) [13]. The authors found that perfect 3′-supplementary interactions can enhance target affinity. Additionally, they demonstrated that the seed and the 3′-supplementary regions can be connected by an unstructured loop of 1–15 nucleotides in the mRNA target. Even when the loop region has base-pairing potential, Ago2 holds a stable conformation that avoids it [13]. Further evidence indicates that the benefit of microRNA 3′-pairing varies depending on the microRNA sequence and the 3′-pairing architecture, thus understanding what features determine the use of 3′-pairing was suggested to require large measurements of multiple microRNA sequences [17]. An explanation of the contradictory findings about the relevance of the 3′-supplementary interaction suggested that the differences found in the repression level between the global analyses and those of specific microRNAs might be due to a “dilution” effect of the few strong interactions among the most frequent weak interactions [18].

In the present study, we evaluate the contribution of the microRNA canonical seed types and the 3′-supplementary regions to target mRNA repression by analyzing all the evolutionary conserved microRNAs [19] and all mRNA transcripts expressed in prostate tissue samples of the prostate adenocarcinoma cohort of The Cancer Genome Atlas (PRAD-TCGA). MicroRNA–mRNA interactions were identified by sole base complementarity, and the regulatory magnitude of the interactions was evaluated by the Spearman correlation coefficient of the expression of both RNA partners in the RNA-seq and small RNA-seq transcriptomes of the PRAD-TCGA. Overall, our investigation suggests that the comparative analysis of RNA-seq transcriptomes of mRNAs and small RNAs of a large number of tissues confirms known microRNA–mRNA interaction determinants and may be useful to produce novel hypotheses.

## 2. Results

### 2.1. Identification of microRNA–mRNA Interaction Sites in the PRAD-TCGA Transcriptome Based on Nucleotide Base Complementarity

Given the tissue specificity of microRNA expression, we performed this study within a single tissue type. We chose the PRAD cohort because it is large and extensively studied for microRNAs [20,21,22,23]; in addition, PRAD is a very heterogeneous cancer, thus biases due to major cancer subtypes are unlikely [24]. We investigated the expression of the conserved human microRNAs (221 out of the 2606 microRNA entries available in Targetscan 7.2) and all the human protein-coding transcripts annotated in Ensembl (20,440) in the 546 samples of the PRAD-TCGA cohort (20,531 genes and 2588 microRNAs expressed) (see pipeline analysis in Figure 1). To increase the signal/noise ratio, we filtered out low-expression transcripts, arbitrarily defined as those undetected in ≥20% of the samples, narrowing the list of microRNAs and mRNA to 143 and 16,786, respectively (Appendix A and Appendix A). Aiming to compare the target efficacy of different microRNA pairing sites, we studied 6 mer, 7 mer-A1, 7 mer-m8, and 8 mer microRNA seeds with or without the 3′-supplementary pairing regions, defined as the four nucleotides at positions 13–16 of the microRNA (hereafter denoted as 6 mer + suppl, 7 mer-A1 + suppl, 7 mer-m8 + suppl, and 8 mer + suppl) (Figure 2A). The choice of the 3′-supplementary pairing site was based on the literature since mounting evidence of its repressive contribution has been published [1,13]. We determined all the mRNA sites with perfect complementarity to these eight types of canonical microRNA pairing sites, presuming they represent putative microRNA binding sites. This analysis was performed independently for the three regions of the mRNAs (5′UTR, CDS, and 3′UTR). The examination of seed sequence complementarity between the 143 microRNAs and the 16,786 mRNAs identified 3,065,731 putative microRNA interaction sites on the mRNAs (285,237 at 5′UTR, 1,226,568 at CDS, and 1,553,926 at 3′UTR). Since each mRNA can bear more than one putative microRNA site for a given microRNA, in subsequent calculations, we only counted the interactions involving microRNAs with a unique site in the mRNAs. Therefore, if an mRNA has more than one site for a given microRNA, these interactions were excluded from the analysis. Yet, if the same mRNA had sites for different microRNAs, these interactions were compiled as long as there were single sites per microRNA. We performed this calculation for each of the three mRNA regions independently. The analysis was iterated for each of the 143 selected microRNAs, obtaining a total of 1,116,787 sites (193,203 at 5′UTR, 443,323 at CDS, and 480,261 at 3′UTR), corresponding to an average microRNA interaction number of 1351 ± 621 at 5′UTR, 3100 ± 748 a CDS, and 3358 ± 602 at 3′UTR (Appendix A). As expected, for the eight interaction types in the three regions studied, the number of predicted interactions decreases with the seed size and the incorporation of the 3′-supplementary regions (Appendix A). The same analysis performed with a random supplementary sequence resulted in 1,224,398 interaction sites (211,226 at 5′UTR, 527,900 at CDS, and 485,272 at 3′UTR).

### 2.2. Validation of microRNA Repressive Mechanisms Using microRNA Activity Inferred from microRNA–mRNA Expression Correlations in PRAD-TCGA

We initially sought to evaluate if the correlation of microRNA–mRNA pairs abundance in the tissues was sufficient to validate established microRNA repression rules using the 143 conserved microRNAs in the pool. Altogether, we found only negative correlations between the microRNA–mRNA pairs defined by base complementarity, regardless of the seed type and gene region (Figure 2B). The finding of an overall negative Z-score of inferred repression suggests that the bioinformatic strategy applied is capable of identifying microRNA activity. The comparison of a Z-score of microRNA–mRNA correlation for each type of seed showed that inferred repression increases with the seed’s length, i.e., the number of nucleotide bases involved in the interaction (6 mer < 7 mer < 8 mer) (Figure 2B and Appendix A, Table 1 for comparisons with *p* < 0.05 in at least one region, and Appendix A for all comparisons), thus confirming the previously reported hierarchy of site efficacy: 8 mer > 7 mer > 6 mer [2,3,16,25,26]. In contrast to the previous findings [2,16,25], the average Z-scores indicate that the 7 mer-A1 seed interaction resulted in significantly more repressive than the 7 mer-m8 in the three regions of the mRNA studied (Figure 2B). Additionally, the 7 mer-m8 shows no significant difference from the 6 mer seed in both UTRs (*p*-value < 0.01) (Figure 2B).

In agreement with the established rules of the microRNA-targeting mechanism, the hierarchy of microRNA–mRNA interactions productivity indicated by the Z-score of the correlations of the comparisons is 3′UTR > CDS > 5′UTR (Figure 2B,C) [2,3,4], supported by the *p*-values of the comparison among the sites in the three different regions (*p*-value < 0.01) (Table 2). This finding provides further evidence of the value of microRNA–mRNA expression correlations in tissues to study microRNA repression characteristics.

The comparison of interactions mediated by sole seeds and seeds plus native 3′-supplementary regions shows that the 3′-supplementary interaction reduces the Z-score of the interactions, hence likely contributing to the correlation’s inferred repression (Figure 2C and Appendix A). This effect is statistically significant for 6 mer seeds at the 3′UTR and the CDS and the 7 mer-m8 seeds at the 3′UTR (*p*-value < 0.01) (Figure 2C and Table 1 and Appendix A for complete list of comparisons). The lack of statistical significance of the differences may be influenced by the reduced number of interactions of the longest seeds (Appendix A and Appendix A). Control of the specificity of the analysis was performed by substituting the 13–16 nt positions of all microRNAs with random sequences differing in at least three nucleotides from their native supplementary site while conserving the GC content. These chimeric microRNAs are expected to disrupt the native base pairing at the mRNA target 3′-supplementary site, hence are not expected to increase the target efficacy of the sole seeds. Concordantly, no significant differences in Z-scores of repressions between these chimeric microRNAs and the sole seed were found (*p*-value < 0.01) (Figure 2C, Appendix A). These observations imply that base pairing between the native 13–16 nt supplementary region and the target mRNA site contributes to the reduction of the Z-score and thus to the inferred target repression.

### 2.3. Insight into the Characteristics of microRNA–mRNA Interaction Involving Beneficial 3′-Supplementary Pairing Regions withdrawn from microRNA–mRNA Correlations

Our previous results suggested that the Z-score of microRNA–mRNA correlations derived from tissue transcriptomes is sensitive to the repression driven by 3′-supplementary 13–16 nt microRNA region, thus we sought to investigate further characteristics of this interaction. For the following analyses, we focused on the seed plus 3′-supplementary sites of 6 mer + suppl interactions at 3′UTR and CDS and 7 mer + suppl interactions at 3′UTR, since they showed a statistically significant increase in the inferred target repression compared to sole seeds using our informatic pipeline (Table 1); therein, we selected the microRNAs with lower Z-score inferred repression on targets bearing 3′-supplementary pairing possibilities than on targets without 3′-supplementary pairing possibilities, in at least one of the three gene regions studied, as determined by a T-test of the Z-score of the differences (*p*-value ≤ 0.05) (Appendix A). Thirty-nine microRNAs resulted in Z-score differences “sole seed − seed + suppl” lower than 0, thus performing more repressively when able to complementarily pair with their targets through the 3′-supplementary sites. These 39 repressive microRNAs have no significant difference in expression (Appendix A) or the number of possible mRNA targets (Appendix A) than the total 143 microRNAs analyzed (*p*-value ≤ 0.05). Meanwhile, 18 microRNAs resulted in Z-score differences greater than 0, thus seeming to attenuate the repression exerted by the sole seed when they interact through mRNA targets via complementary pairing at the 3′-supplementary sites. If each region is analyzed separately, the 3′-supplementary sites that follow or oppose a repressive contribution to their sole seeds are comparatively 19 vs. 11 of 6 mer (*p*-value = 0.2 binomial test of significance) and 16 vs. 4 of 7 mer-m8 at the 3′UTR (*p*-value = 0.01 binomial test of significance), while 21 vs. 8 of 6 mer was observed at the CDS (*p*-value = 0.02 binomial test of significance). The higher proportion of microRNAs exerting Z-score-inferred repression when able to pair with their targets through the 13–16 nt 3′-supplementary site suggests that this bioinformatic approach is sensitive to the 3′-supplementary region repressive activity of individual microRNAs.

The contribution to microRNA target efficacy accomplished by the bridge between the 3′-supplementary sites at 13–16 and the seed has also been the subject of investigation. Crystallographic studies of Ago2 engaged with a target RNA showed that the seed and the 3′-supplementary regions could be bridged by an unstructured target loop of 1–15 nucleotides when complexed with AGO2, even when central complementary bases are available for pairing [13]. Moreover, 8 mer + suppl interactions with bridges up to 10 nucleotides long on the mRNA were more repressive than 8 mer sole seeds. The same study proposes that interactions involving 3′-supplementary regions with high GC content can be established using bridges of up to 15 nt [13]. Since the bridge between the seed and the 3′-supplementary region of the microRNA has five nucleotides for the 6 mer + suppl (8–12 nt) and four nucleotides for 7 mer-m8 + suppl (9–12 nt), the same length at the mRNA bridge means no loop formation in any of the two molecules and is defined as “zero offset” (Figure 3A). Longer and shorter mRNA bridges imply a loop formation in the mRNA (positive offset values) or microRNA (negative offset values). Seeking to assess if the Z-score of microRNA repression withdrawn from the PRAD-TCGA tissue data was influenced by the offset of the microRNA–mRNA interaction, we scored the interactions yielding Z-score differences “sole seed − seed + suppl” lower than 0, computing the offsets of the predicted interactions with their potential mRNA targets (defined as above, i.e., the ability to pair to the mRNA target using the 13–15 nt microRNA site). An offset of +10 was set as the maximum, involving mRNA bridges of 15 nt for 6 mer seeds and 14 nt for 7 mer-m8 seeds. As a control, we performed the same analysis for the 143 microRNAs and chimeras generated by random nucleotide substitution with at least three nucleotides divergent at the 13–16 nt site, conserving the GC content of the native site (as performed previously for the different type of microRNA–mRNA seed-3′ supplementary interactions). As expected, the majority of Z-scores seed vs. seed-3′supplementary interaction differences are independent of the offset of the interaction for the 143 total microRNAs, regardless of the seed type (6 mer or 7 merm8) or gene location analyzed (CDS or 3’UTR); the few deviated differences (e.g., 6 mer offset + 2) produce Z-scores values above −0.1 to −0.2 (Figure 3B). Meanwhile, the 3′-supplementary interactions of the 39 repressive microRNAs generate heterogeneous Z-scores depending on the offset (Figure 3B). For the 6 mer interactions occurring at the 3′UTR, an increase in repression is found at shorter microRNA loops (negative offsets), reaching a maximum of 0 offsets (Z-scores of approximately −0.4). Furthermore, interactions involving mRNA loops (offsets + 1 to +10) produce mostly repressive offsets, and the +1 offset is one of the most repressive lengths in the three types of interactions analyzed (Z-scores between −0.4 and −0.6). A similar pattern of repression is depicted for the two other datasets (6 mer CDS and 7 mer-m8 3′UTR), though the differences are less significant, perhaps due to the smaller number of interactions comprising them. Again, a control evaluation of sequence specificity of the interactions using chimeric microRNAs built by substituting the 39 native microRNA sequences with random nucleotides at positions 13–16 abolish target-pairing possibilities at these sites (different in at least three positions while conserving the native GC content). The latter analysis did not yield significant Z-score differences with the sole seed at any offset (*p*-value < 0.01) (Figure 3B, repressive (39), random). Concordantly, an identical analysis using 39 non-repressive microRNAs randomly sampled from the 104 non-repressive microRNAs (143–39) did not yield significant Z-score differences relative to sole seeds (Figure 3B, non-repressive (39), native). The higher Z-score variation observed in the 39 microRNAs compared to the 143 microRNAs analyses seems to derive from the smaller sample size and displays similar variation. Overall, the controls support the specificity of the approach.

To add up the information from the three datasets interrogated, we scaled their Z-scores using minimum–maximum scaling to normalize their different Z-score ranges, and then we combined the three values (Appendix A). This integrated analysis suggests that smaller microRNA loops are more repressive, while a one-nucleotide loop in the mRNA (+1 offset) provides one optimum repression (Appendix A). In addition, other positive offsets (+3 and others) withstand Z-scores similar to or lower than +1, suggesting that longer loops in the mRNA bridge may also contribute to 3′supplementary site-mediated microRNA repression.

Since base content bias has been previously proposed at the 3′-supplementary microRNA interaction sites [13], we analyzed the GC content of the microRNAs with productive repressive 3′-supplementary pairing in comparison with the total microRNAs. A notorious decrease in the GC content in the stretch between positions 13–16 of the microRNAs with a repressive 3′-supplementary site is observed for the three types of interactions analyzed (Figure 4A). In addition, positions 5 and 19 of 6 mer 3′UTR sites have a GC content deviated from the total. The base composition analysis per position shows a preference for base A at positions 13–15 of the 3′-supplementary pairing region of the microRNA and for base U at position 16 for the three interaction types analyzed (Figure 4B–E). The enrichment of bases U and A observed at positions 1 and 2 of all the microRNAs (Figure 4B) has been previously described in mammals and other species [16,27,28]. Overall, microRNAs that establish repressive 3′-supplementary interactions with their targets seem to have low GC content at positions 13–16 nt. Although the 6 mer analysis suggests additional compositional skews, more interactions would be needed to confirm these observations.

### 2.4. An Executable Tool to Identify the microRNA–mRNA Seed + Suppl Interactions

We developed two Windows-compiled executable scripts to search for seed and seed + suppl interactions in human genes solely based on mRNA–microRNA perfect base-pairing potential. Through the usage of python regular expressions, both algorithms identify 6–8 mer seeds, and 3′-supplementary interactions intervened by 1–15 nucleotides between the seed and the 3′-supplementary pairing regions. Each algorithm searches for complementarity to a given microRNA inside all transcript sequences available in the human genome and the other searches for all microRNA sites in a given mRNA transcript. Therefore, the first tool retrieves all the mRNA targets for an input microRNA, and the other retrieves all microRNAs that interact with an input mRNA transcript. The output file includes microRNA and mRNA identity as well as the specific nucleotides involved in the interaction, the nucleotide bridge length, and the nucleotide bridge sequence. The search discriminates CDS and 3′ and 5′ UTR, and all annotated transcript variants. The source code in Python and two windows executable files, the manual, and the files are available on GitHub (https://github.com/JuanTrinidad/microRNA-supplementary-interaction).

## 3. Discussion

The principles of microRNA efficacy over target mRNAs have been established by several experimental and in silico strategies. Among several empirical approaches, immunoprecipitation of AGO-complexes, in vitro affinity determinations, reporter gene assays at low and high-throughput levels, structural studies, and correlations between microRNA and mRNA abundance in cells have been assessed. The latter approach was applied in settings involving the enforced modulation of the microRNA abundance by transfection or in knockouts of cell lines and provided most of the current understanding of microRNA action on its targets [2,3,4,7,8], but has the disadvantage of being limited to several microRNAs. Meanwhile, evidence of microRNA regulation in intact cells or tissues has also been withdrawn from the correlation between microRNA–mRNA pairs’ abundance, but this was usually used as a criterion to support the microRNA regulation on a single or a few mRNA targets [29,30,31,32,33]. Since Cancer Genome projects involving multi-omics studies have provided matched RNAseq and small-RNAseq of hundreds of tissues, we reasoned they would allow the high-throughput transcriptome-wide assessment of microRNA–mRNA expression correlations in human tissues.

Here we sought to investigate if the analysis of microRNA–mRNA correlations in large tissue datasets and the aggregation of information from multiple microRNAs can validate established microRNA–mRNA pairing rules, which may indicate its putative potential to expose novel features of their interaction. We thus performed a transcriptome-wide approach to measure the correlation between all the mRNA transcripts and the conserved microRNAs expressed in the PRAD-TCGA cohort. Our strategy to assign microRNA –mRNA pairs is solely based on perfect Watson–Crick base complementarity at the 2–7 or 2–8 (6 mer and 7 mer-8 mer microRNA seeds, respectively) and the 13–16 (microRNA 3′-supplementary region) positions of the microRNA. Although this sole criterion is permissive, it has the advantage of reducing potential biases of the microRNA target prediction algorithms (target site conservation, the free energy of the interaction, site accessibility, target-site abundance, local AU content, GU wobble pairing in the seed, exact position in the transcript, length of the transcript regions, and machine learning features) [34]. In support of this notion, it has been stated that a simple text search yields more reliable predictions than some common algorithms [1]. Furthermore, the use of this single criterion maximizes the number of interactions that can be interrogated. Another advantage of our approach is the use of endogenous-level transcripts since concerns have been raised about artifactual observations due to the non-physiological stoichiometry of the perturbation-based approaches [18]. Meanwhile, our approach has limitations. First, the lack of requirement of standard microRNA–mRNA target prediction rules may compute non-physiological interactions. To counteract this liability, we only examined conserved microRNAs, which better comply with established microRNA–target interaction hallmarks. Second, although the transcriptomic data of the PRAD-TCGA are large, it is still limited, particularly when the interactions are to be classified into exclusive categories (seed class, supplementary region, and offset length). Third, the abundance of microRNAs and mRNAs in the cells is determined by multiple factors, including evolutionary constraints, transcriptional and post-transcriptional gene regulation, and stoichiometry, while the impact of microRNA repression on mRNA steady-state levels is expected to reach an average maximum near 10%, as seen in microRNA perturbation in vitro experiments [7,35]. Fourth, microRNAs can repress mRNA translation without modifying its abundance, a regulatory step that is not captured by our study. Fifth, not all known microRNA–mRNA interactions were included in the analysis, such as seed–site interactions with mismatches and compensatory supplementary sites, or UTRs bearing multiple seed sites for a given microRNA. Additionally, UTRs regulated by multiple microRNAs are informatically processed equally for each microRNA, although their individual actions are likely diverse. Altogether, these shortcomings may increase the noise of the analyses, hence reducing the biological meaning of the findings.

Despite the above-mentioned limitations, the analysis of Z-scores of microRNA–mRNA correlations in PRAD-TCGA allowed the confirmation of established rules or microRNA target regulation, such as the overall negative correlation between microRNAs and predicted mRNA target abundance, the hierarchy of repression dependent on the gene region (3′UTR > CDS > 5′UTR) and the seed length (6 mer < 7 mer < 8 mer), and the contribution of the 3′-supplementary pairing at 13–16 nt of the microRNA repression [1,2,5,13,17,19,36]. It is worth noting that, in terms of Z-score-inferred repression, the 6 mer + suppl is not significantly different from the 7 mer-A1 interaction, which raises awareness of the possibly underestimated relevance of the 6 mer + suppl interaction in vivo. Although the 6 mer site per se is known to be less specific for repression than longer seeds and is classified as a “marginal site” [2,19], the addition of a 3′-supplementary four-nucleotide base pairing (microRNA position 13–16) is known to contribute to its target efficacy, and our approach suggests that this contribution may be biologically relevant. Likewise, the mapping of microRNA–mRNA interactions in live neural tissues found a higher preponderance of 6 mer sites than in vitro approaches [5]. Additionally, and in contrast to previous reports [2,25,26], our data suggest that the 7 mer-A1 seed is significantly more repressive than the 7 mer-m8 in the three transcript regions analyzed. Nevertheless, it has been proposed that the contribution of the 7 mer-m8 and the 7 mer-A1 can substantially differ for different microRNAs [25]. Further investigation is needed to understand if this unexpected finding is due to the nature of the models studied, the bioinformatic pipelines used, or another source of variation.

The functionality of the 3′-supplementary pairing in microRNA–mRNA interactions was described several years ago [2], and further insight into its molecular basis was recently achieved [13,17,18,37]. Unlike the first studies that used a 2–6 nt seed-supplementary bridge length [2], recent studies showed that up to 15-nt-long loops on the target mRNA could bridge the seed and the 3′-supplementary regions [13,17]. The discrepancies in the relative contribution of the 3′-supplementary pairing to the target repression among the aforementioned studies have been ascribed to the different methodological strategies of the analyses [5,18]. In this context, we thought that the study of a large number of interactions in tissue samples would contribute to the understanding of characteristics of the 3′-supplementary interactions, such as the bridge length, base preferences, or repression efficacy.

Our correlation-based approach found that 3′-supplementary nucleotide pairing significantly enhances the repressive activity of 6 mer and 7 mer-m8 seeds, while a tendency is also observed for the 7 merA1 and the 8 mer seeds at the CDS and 3′ UTR (Appendix A). Individually, at least 39 conserved microRNAs may benefit from the 3′ supplementary site for repression when the target sites are located at the 3’UTR and CDS of mRNAs targets. In addition, one of the more repressive offsets occurs at +1, which has also been identified as an optimal offset in previous reports [13,17]. Moreover, the pattern of repression for offsets larger than +1 suggests that the 3′-supplementary pairing can be equally repressive at higher offsets, such as +3 or +7. The heterogeneous 3′ microRNA architecture of the canonical seeds may explain the heterogeneity of the repression per offset, as has been recently described for 3’compensatory microRNA–miRNA interactions led by imperfect seeds matches [17]. Interestingly, the two binding modes recently reported for compensatory interactions by McGeary et al., 2022 (offset 0–1 and +3–4) may be connected to our findings of a repressive offset at offsets larger than +1 [17].

Our study led to the identification of sequence patterns enriched in the 39 repressive supplementary microRNAs, including a low GC content with a higher adenine proportion at the 3′-supplementary regions, whose relevance must be validated by additional methods. In opposition, a low GC content in this region was previously associated with a lower affinity of the interaction evaluated in vitro using a specific microRNA–RNA target pair [13,17,18]; this discrepancy may be due to the different microRNAs analyzed in each study, the method used to study the interactions (in vitro tripartite complex affinity or pull-down, reporter gene assays, or exogenous modulation of individual microRNAs), or the setting of the study (in a tube, in cells, or in tissue), but also still unknown factors and the specific constraints of each study. Indeed, Moore et al.’s CLEAR strategy performed in vivo in live brain tissue (AGO-HITS-CLIP with a covalent ligation of endogenous AGO-bound RNAs) suggested that 3′ auxiliary pairing of microRNA may be more miRNA-specific and tolerant to the diversity of patterns than generally assumed [5], a notion that was also suggested by high-throughput studies of AGO2 binding [38].

The tool developed in the present study differs from the existing ones because it detects miRNA–mRNA interactions solely defined by perfect base pairing complementarity, without any further requirement. Likewise, the supplementary pairing region is restricted to the 13–16 nt position of the microRNA, the interaction sites, and the offset of the interactions reported, as well as the 6 mer seed, all conditions that are not often computed by currently available microRNA–target analysis tools.

Our study represents a new high-throughput strategy to investigate the microRNA–mRNA target interaction by withdrawing patterns from transcriptome-wide expression correlations in large datasets of tissues, representing an unperturbed biological set. The findings confirm existing knowledge, supporting the sensitivity of the approach and the potential biological meaning of the outcomes, which may contribute to investigating novel aspects of the microRNA–mRNA interaction.

## 4. Materials and Methods

### 4.1. MicroRNAs and mRNAs Transcriptomic Data

RNAseq and small RNA-seq data of the TCGA-PRAD (normal and tumor conditions) were obtained from Firebrowse (firebrowse.org, Massachusetts Institute of Technology&Harvard, Cambridge, MA, USA, version 1.1.40). The microRNA conservation scores were obtained from the TargetScan 7.2 database (targetscan.org, Massachusetts Institute of Technology, Cambridge, MA, USA, version 7.2) [19], and the 221 microRNAs with the highest “Family Conservation score” (score 2 defined by TargetScan), hereafter denoted as “conserved microRNAs”, were selected for this study.

Only microRNAs and mRNAs detected in at least 80% of the samples were included in the analysis (Appendix A and Appendix A, see complete pipeline analysis in Figure 1).

### 4.2. Inference of microRNA Activity from microRNA–mRNA Expression Correlation in the PRAD-TCGA Transcriptomes

The availability of matched RNAseq and small-RNAseq and transcriptomes of hundreds of tissues due to global cancer initiatives allows the high-throughput transcriptome-wide assessment of microRNA–mRNA correlations. Our approach evaluates the efficacy of microRNA–mRNA interactions using the Spearman correlation coefficient of the expression of both molecules calculated from the RNAseq data of the 546 patient tissues (normal and tumor conditions) of the PRAD-TCGA cohort (Figure 1). We include positive and negative correlations between the mRNA–microRNA pairs. Additionally, in the absence of consensus about the threshold value of the significant biological correlation expected in tissues for direct microRNA repression, we analyzed all the correlations regardless of their magnitude and sign.

### 4.3. Canonical Seeds and 3′-Supplementary microRNA–mRNA Interactions Identification

Mature human microRNA sequences were obtained from the TargetScan 7.2 database [19]. Human mRNA sequences of all protein-coding genes were obtained using BioMart (Ensembl GRCh38.13).

An in-house script written in Python programming language was developed to identify all the sequences complementary to all the canonical microRNA seeds (as described by TargetScan) of the 143 microRNAs (script available on GitHub, https://github.com/JuanTrinidad/microRNA-supplementary-interaction). The TargetScan nomenclature was used to name the microRNA–mRNA interaction types, which comprise 6 mer, 7 mer-A1, 7 mer-m8, and 8 mer microRNA seeds (Figure 2A). For this study, the 3′-supplementary region of an mRNA–microRNA interaction refers to microRNA nucleotides at positions 13–16 [2,13] (Figure 2A). For both the sole seed and seed plus 3′-supplementary region, only perfect complementary interactions with the mRNA were computed. A variable-length bridge of unpaired nucleotides can separate these two sequence modules. We investigated mRNA loops with a maximum length of 15 nucleotides based on previous functional studies [13]. The difference in the length of the microRNA and the mRNA unpaired bridges is referred to as the “offset” of the interaction involving 3′-supplementary sites.

We identified the eight types of putative microRNA–mRNA interactions described using Python regular expressions, comprising 6 mer, 7 mer-A1, 7 mer-m8, 8 mer, 6 mer + suppl 7 mer-A1 + suppl, 7 mer-m8 + suppl, and 8 mer + suppl (Figure 2A). We chose the transcript variant with the highest number of putative microRNA interactions for each gene. In addition, as per each microRNA, we selected the mRNAs that presented only one predicted microRNA–mRNA interaction, pondering the CDS, 3′UTR, and 5′UTR independently. As a result, the computed mRNAs only have a single microRNA site at the region considered for every given microRNA analyzed, thus avoiding the co-occurrence of the sites.

Controls were performed to assess the sequence specificity of the supplementary interaction. For that purpose, chimeric microRNAs were generated substituting the native supplementary sequence of the microRNAs with a random sequence (Appendix A). To maximize the dissimilarity between the native and the random supplementary sequences, a Levenshtein distance ≥ 3 was set. The GC content of the random sequence was kept identical to the native sequence to disrupt the pairing potential to the target without modifying the base composition of the sequence. A random supplementary sequence was assigned to each microRNA, and the microRNAs with the same native supplementary sequence were assigned the same random supplementary sequence.

### 4.4. MicroRNA–mRNA Pairs Correlation Analysis

For the selected list of microRNA and mRNAs, the normalized Spearman correlation coefficient (Z-score centered on microRNA) of the abundance of all putative microRNA–mRNA interacting pairs were calculated with the Python Pandas library [39] using TCGA RNA-seq (mRNAs) and small RNA-seq (microRNAs) normalized data (546 tissue samples) obtained from FireBrowse (firebrowse.org).

### 4.5. Statistical Analysis

Unless specified, statistical differences were assessed by a two-tailed T-test, and the *p*-values were adjusted by Bonferroni correction using SciPy library version 1.6.2. The Fisher exact test was performed using SciPy library version 1.6.2. In all cases, adjusted *p*-values were analyzed. All the plots were generated using the Seaborn library 0.11.2.

### 4.6. Executable Files

The executable files used to search for microRNA–mRNA interaction sites were developed using PySimpleGUI version 4.55.1 graphical user interface and PyInstaller version 4.10 (pysimplegui.org, PySimpleGUI Tech LLC) and are available on GitHub (https://github.com/JuanTrinidad/microRNA-supplementary-interaction).

## Figures and Tables

**Figure 1 ncrna-09-00015-f001:**
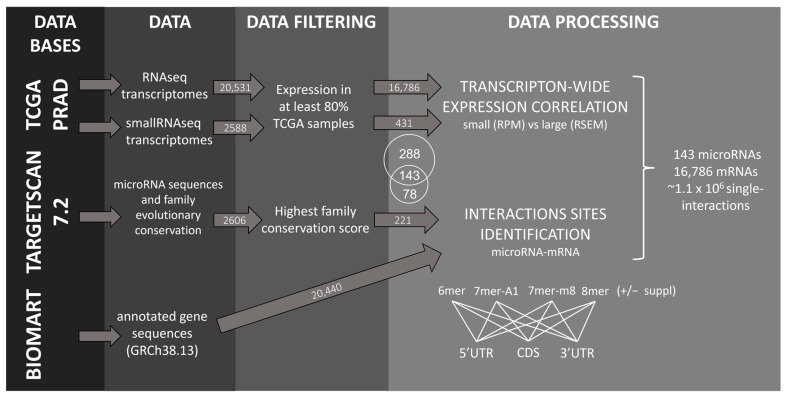
Data analysis pipeline.

**Figure 2 ncrna-09-00015-f002:**
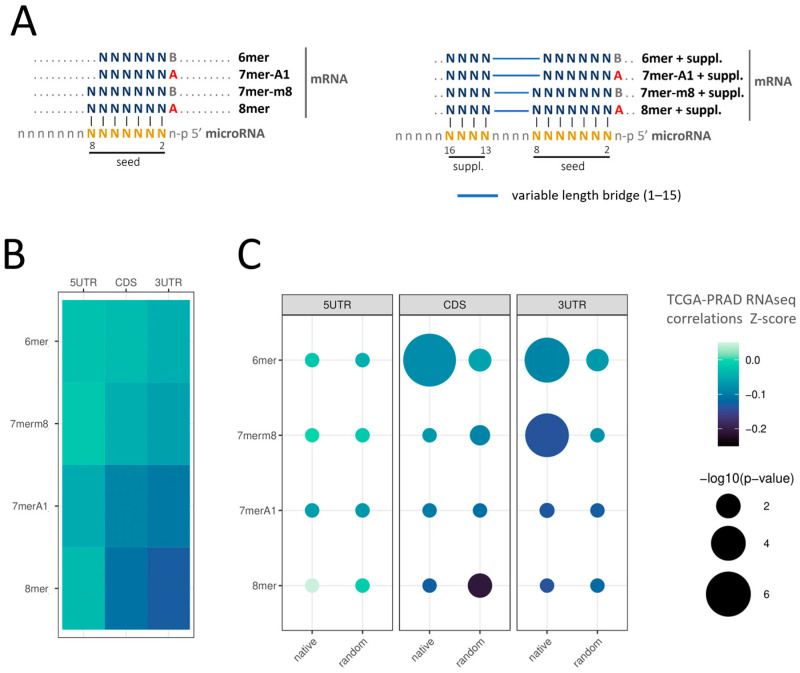
Repression of mRNA–target interactions with different pairing sites. (**A**) MicroRNA seed (6 mer, 7 mer-A1, 7 mer-m8, and 8 mer) and supplementary pairing sites (positions 13–16 of the microRNA). A in the sequences corresponds to Adenine and B corresponds to the other three nucleotides except for Adenine. (**B**) Heatmap of the average Z-score correlations between the 143 conserved microRNAs and their predicted target mRNAs using sole seed, discriminating the 5′UTR, CDS, and 3′UTR regions. (**C**) Circle matrix of the average Z-score correlations between the 143 conserved microRNAs and their predicted target mRNAs using native or random seed + supplementary interactions. Color scale represents the Z-score of microRNA–mRNA correlations while the diameter of the circles represents the log10(*p*-value) of the comparison between seed + supplementary with the sole seed interactions. −log 10 values of 2, 4, and 6 correspond to *p*-values of 0.001, 0.0001, and 0.000001, respectively.

**Figure 3 ncrna-09-00015-f003:**
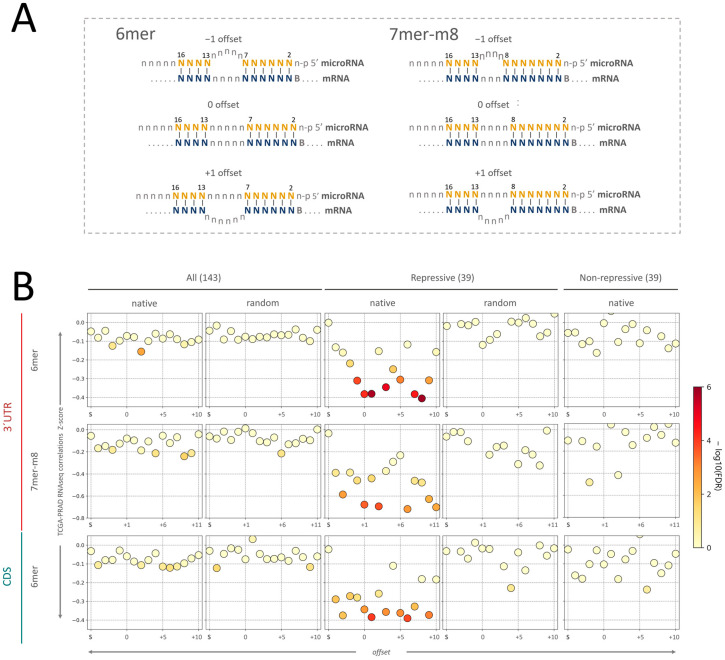
Contribution of the mRNA bridge length to the microRNA seed + 3′-supplementary site mediated repression. (**A**) Graphic representation of the architecture of the microRNA–mRNA interaction using 3′pairing. The bridge region is defined as the unpaired nucleotides from the end of the seed to the beginning of the 3′-supplementary region. Since microRNA–mRNA 3′-supplementary interaction occurs at positions 13–16 of the microRNA, 6 mer and 7 mer-m8 seeds have 5 and 4 nt microRNA bridges, respectively; thus, these mRNA bridge lengths mean no loop formation for the respective seeds and are denoted as “zero offset”. Positive offset values imply loop formation in the mRNA and negative offset values in the microRNA. (**B**) Analysis of bridge length effect on target repression using 3′-supplementary sites (Z-transformed score). The TCGA transcriptomes and the 143 total microRNAs included in this study as a control, the 39 microRNAs with repressive 3′-supplementary interactions using the native sequence and a random sequence (positions 13–16 of the microRNA), and a random group of 39 microRNAs with non-repressive 3′-supplementary sites conserving GC content of the native site. The region of the genes and the type of seeds analyzed are indicated in the left margin. The abscise axis indicates the offset length. The −log10(*p*-values) of the differences between each seed + suppl length and the respective sole seed interactions are indicated by the color intensity bar. S stands for “seed” and indicates the Z-score of sole seed interactions.

**Figure 4 ncrna-09-00015-f004:**
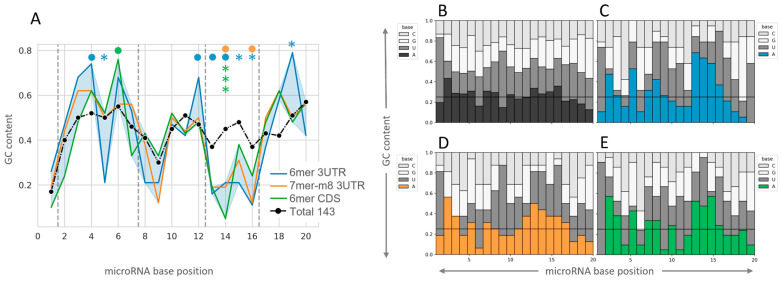
Nucleotide composition of the microRNAs with repressive 3′-supplementary interactions. (**A**) Distribution of GC content along the positions of the microRNAs with the three repressive 3′-supplementary interactions indicated (colored) vs. the whole set of microRNAs analyzed (black). Fisher exact test was performed to compare each of the represented sites with the total 143 microRNAs (● < 0.1, * < 0.05, *** < 0.001). (**B**–**E**) Base composition per microRNA nucleotide position. Seed types are represented by colors as indicated in (**A**) (**B**: All microRNAs, **C**: Blue/6 mer 3′UTR, **D**: Orange/7 mer-m8 3′UTR, and E: Green/6 mer CDS). Horizontal lines represent the ¼ frequency expected for a random nucleotide distribution.

**Table 1 ncrna-09-00015-t001:** Comparison of Z-scores of microRNA–mRNA correlation classified by interaction type.

Group1	Group2	5’UTR	CDS	3’UTR
6 mer	7 mer-m8	n.s.	6.09 × 10^−3^	n.s.
6 mer	7 mer-A1	1.14 × 10^−2^	2.36 × 10^−42^	4.82 × 10^−45^
6 mer	8 mer	n.s.	1.74 × 10^−25^	9.23 × 10^−35^
7 mer-m8	8 mer	n.s.	2.24 × 10^−13^	4.41 × 10^−23^
7 mer-A1	7 mer-m8	2.93 × 10^−3^	3.94 × 10^−14^	9.48 × 10^−21^
7 mer-A1	8 mer	n.s.	n.s.	1.24 × 10^−2^

6 mer	6 mer suppl	n.s.	2.84 × 10^−8^	9.9 × 10^−7^
6 mer	7 mer-m8 suppl	n.s.	n.s.	3.66 × 10^−8^
6 mer	7 mer-A1 suppl	n.s.	1.71 × 10^−4^	2.48 × 10^−9^
6 mer	8 mer suppl	n.s.	1.2 × 10^−2^	3.72 × 10^−3^
7 mer-m8	7 mer-m8 suppl	n.s.	n.s.	1.74 × 10^−6^
7 mer-m8	8 mer suppl	n.s.	n.s.	1.52 × 10^−2^
6 mer suppl	7 mer-m8	n.s.	2.84 × 10^−3^	6.92 × 10^−4^
6 mer suppl	8 mer	n.s.	n.s.	1.74 × 10^−3^
7 mer-A1 suppl	7 mer-m8	n.s.	2.20 × 10^−2^	2.43 × 10^−7^

**Table 2 ncrna-09-00015-t002:** Comparison of Z-scores of microRNA–mRNA correlation classified by gene region.

Group1	Group2	6 mer	7 mer-A1	7 mer-m8	8 mer
3UTR	5UTR	5.98 × 10^−9^	2.69 × 10^−14^	4.34 × 10^−7^	7.30 × 10^−12^
3UTR	CDS	2.62 × 10^−8^	1.37 × 10^−2^	n.s.	n.s.
5UTR	CDS	n.s.	6.25 × 10^−8^	6.82 × 10^−5^	2.06 × 10^−7^
**Group1**	**Group2**	**6 mer suppl**	**7 mer-A1 suppl**	**7 mer-m8 suppl**	**8 mer suppl**
3UTR	5UTR	2.81 × 10^−6^	3.62 × 10^−2^	1.14 × 10^−5^	1.29 × 10^−3^
3UTR	CDS	n.s.	n.s.	5.11 × 10^−3^	n.s.
5UTR	CDS	7.30 × 10^−5^	n.s.	n.s.	4.49 × 10^−3^

## Data Availability

PRAD-TCGA RNA-Seq and miR-Seq data is available at Firebrowse.org; The scripts and software developed in this study are available at Github.com/JuanTrinidad/microRNA-supplementary-interaction.

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
