# Peer review of "Transcriptome-Wide Analysis of microRNA–mRNA Correlations in Tissue Identifies microRNA Targeting Determinants"

_ncrna, 2023, doi:10.3390/ncrna9010015_

Round 1

Reviewer 1 Report

This manuscript presented a bioinformatic method to investigate the repression contribution of microRNA elements over target mRNAs via transcriptome-wide analysis.  Four different k-mer types of the seed region, 3’-supplementary regions, and the bridge offset between seed and 3’-supplementary regions are investigated. The organization and presentation of the manuscript are clear, but some issues need to be fixed.

Major comments:

1.     Some of the numbers in Figure 1 are not explained either in the main text or the caption, which makes them hard to interpret. For example, what is 228 in the Venn Diagram, and what is 431 above the Venn Diagram? Note that 431 != 228 + 143.  

2.     The resolution of some figures is too low. For example, it is hard to see the labels in Supplementary Figure 3. Also, in Figure 4, the label of y-axis in panel B—E should not be “GC frequency”.

3.     The GitHub link for the software is not available.

Minor comments:

Some typos need to be fixed.

a.     Line 27: “than” should be “as”.

b.     Line 61: “They” should be “The”.

c.     Line 159: “y” should be “and”.

d.     Line 362: “us” should be “as”.

e.     Line 380: “suggest” should be “suggests”.

Author Response

This manuscript presented a bioinformatic method to investigate the repression contribution of microRNA elements over target mRNAs via transcriptome-wide analysis.  Four different k-mer types of the seed region, 3’-supplementary regions, and the bridge offset between seed and 3’-supplementary regions are investigated. The organization and presentation of the manuscript are clear, but some issues need to be fixed.

Major comments:

  1. Some of the numbers in Figure 1 are not explained either in the main text or the caption, which makes them hard to interpret. For example, what is 228 in the Venn Diagram, and what is 431 above the Venn Diagram? Note that 431 != 228 + 143.  

Thank you for noticing. Although most of the numbers in Figure 1 are explained in the text, two numbers remained unexplained, one is 2606, which refers to the microRNA entries of Targetscan 7.2, and the other is 2588, which is the number of microRNAs expressed in PRAD-TCGA. We have now explained the meaning of these two numbers in the text of Result section.

The Reviewer is also right about the error in the Venn diagram of Figure 1, so we thank him/her for noticing the mistaken number. We made a typing mistake, substituting an 8 with a 2. It is 288, not 228, then 431= 288 + 143. The Figure and text are now amended.

  1. The resolution of some figures is too low. For example, it is hard to see the labels in Supplementary Figure 3. Also, in Figure 4, the label of y-axis in panel B—E should not be “GC frequency”.

The Reviewer is right again, the file lost resolution upon compression. We now uploaded a high-resolution Supplementary File. In addition, as suggested we replaced “GC frequency” with “GC content” in Figure 4 axis, which was indeed how we referred to it in the text. We appreciate the recommendation.

  1. The GitHub link for the software is not available.

Minor comments:

Some typos need to be fixed.

  1. Line 27: “than” should be “as”.
  2. Line 61: “They” should be “The”.
  3. Line 159: “y” should be “and”.
  4. Line 362: “us” should be “as”.
  5. Line 380: “suggest” should be “suggests”.

We truly appreciate the text revision. All Reviewer´s pointed errors have been corrected and a few others too.

Reviewer 2 Report

The authors in the manuscript titled "Transcriptome-wide analysis of microRNA-mRNA correlations in tissue identifies microRNA targeting determinants" using the Z-score of correlation as a surrogate marker of microRNA target efficacy, predict the repression of microRNAs on their targets, and develop a bioinformatic tool to predict microRNA-mRNA candidate interactions. The manuscript appears technically sound. however, some points should be addressed before being accepted for publication.

1. Several software has been developed and widely used to identify  microRNA-mRNA candidate interactions.  Is these any advantage in your bioinformatic tool compared with those methods? 

2. Only use normalized Spearman correlation seems not sufficient to assess of the microRNA-mRNA canonical interaction efficacy. Please explain it.

3. Is there any validation experiment involved in this study? Experiment validation is needed to support your method.

4. Please carefully check the formation of references, such as reference 11.

5. Ten keywords seem too much to a research.

Author Response

The authors in the manuscript titled "Transcriptome-wide analysis of microRNA-mRNA correlations in tissue identifies microRNA targeting determinants" using the Z-score of correlation as a surrogate marker of microRNA target efficacy, predict the repression of microRNAs on their targets, and develop a bioinformatic tool to predict microRNA-mRNA candidate interactions. The manuscript appears technically sound. however, some points should be addressed before being accepted for publication.

  1. Several software has been developed and widely used to identify  microRNA-mRNA candidate interactions.  Is these any advantage in your bioinformatic tool compared with those methods? 

We truly appreciate the Reviewer's suggestion since we realize it is relevant to explain this aspect of the software. We now added the following paragraph clarifying the contribution of the tool to the existing resources:

“The tool developed in the present study differs from the existing ones because it detects miRNA-mRNA interactions solely defined by perfect base pairing complementarity, without any further requirement. Likewise, the supplementary pairing region is restricted to the 13-16nt position of the microRNA, the interaction sites, and the offset of the interactions are reported, and 6mer seed are reported, all conditions that are not often computed by currently available microRNA-target analysis tools.”

  1. Only use normalized Spearman correlation seems not sufficient to assess of the microRNA-mRNA canonical interaction efficacy. Please explain it.

We agree with the Reviewer, and we appreciate the choice of the verb “seems” because our results show that the normalized Spearman correlations are sufficient to confirm fundamental microRNA-mRNA targeting hallmarks. That surprised us in fact, in view of the multiple criteria of microRNA-mRNA interaction that had been described. For that reason, we think this is the main finding of our manuscript, which we consider an important contribution to the field that is worth communicating per se. In other words, it has not been shown yet that the bulk study of microRNA-mRNA correlations in tissues, i.e., in unperturbed physiological settings, is sensitive to microRNA targeting rules. While looking at individual microRNA-mRNA interactions, although tissue correlations per se are not likely sufficient to predict all of them, it is likely sufficient for many of them, which is demonstrated by the statical significance of the bulk comparisons between gene location, seed, offset, and supplementary interactions uncovered by our study. Yet, our study does not focus on individual interactions but on global microRNA-mRNA targeting rules inferred from tissue correlations. On the other hand, while general rules apply to most interactions, many biologically relevant interactions do not comply with them (either for being less conserved or for the heterogeneity of microRNA-mRNA interactions), so we reasoned that it is worth incorporating as much information as possible into their study. The tissue correlation constitutes a relevant criterion that may be fundamental for supporting the latter type of interaction.

In agreement with the Reviewer's concern and to make our study self-critical and transparent for the Readers, the originally submitted manuscript includes an extensive paragraph referring to the study's limitations (Discussion, lines 356-374).

  1. Is there any validation experiment involved in this study? Experiment validation is needed to support your method.

We understand the Reviewer's comment. Nevertheless, in our view, there are multiple validations of the study in the literature since the correlations between microRNA-mRNA interactions in the tissues revealed by our study confirm well-established rules of microRNA interactions (including even specific rules of supplementary pairing repressiveness), both at the general and the individual level, and that is the source of the bulk resulting conclusions.  However, we agree that unexpected findings, such as the low GC content of the supplementary interaction and the relative efficacy of the 7mer seeds need to be proven experimentally. In line with the Reviewers comment the originally submitted manuscript includes the following sentence: “Further investigation is needed to understand if this unexpected finding is due to the nature of the models studied, the bioinformatic pipelines used, or another source variation. (Line 293-295)”. Also, lines 414-417 of the Discussion section state “Our study led to the identification of sequence patterns enriched in the 39 repressive supplementary microRNAs, including a low GC content with a higher adenine proportion at the 3’-supplementary regions, whose relevance must be validated by additional methods.”

The major challenge of the experimental validation would be the need for a tremendous effort since it seems to depend on the study of a large number of interactions in cells or organoids to mimic tissue context. In our opinion, that would represent a separate manuscript that is out of the scope of our present communication.

  1. Please carefully check the formation of references, such as reference 11.

We appreciate the indication. We have reviewed the reference and amended Reference 11 and 2 others which had format errors.

  1. Ten keywords seem too much to a research.

We are thankful for the criticism. The number of keywords has been reduced to comply with the suggestion.

Round 2

Reviewer 2 Report

The details of the algorithm of the method should be added.

Author Response

In order to attend to the Reviewer request, we have now added an .exe Phyton source code for the two executable files on the existing Github space as a file named "seeds_plus_supplementary_search_library.py" and modified the manuscript text accordingly (“source code in Python” in Section 2.4 of Results). In addition, the explanation of the rationale of the software was extended.